# Aging-Related Disorders and Mitochondrial Dysfunction: A Critical Review for Prospect Mitoprotective Strategies Based on Mitochondrial Nutrient Mixtures

**DOI:** 10.3390/ijms21197060

**Published:** 2020-09-25

**Authors:** Giovanni Pagano, Federico V. Pallardó, Alex Lyakhovich, Luca Tiano, Maria Rosa Fittipaldi, Maria Toscanesi, Marco Trifuoggi

**Affiliations:** 1Department of Chemical Sciences, Federico II Naples University, I-80126 Naples, Italy; maria.toscanesi@unina.it (M.T.); marco.trifuoggi@unina.it (M.T.); 2Department of Physiology, Faculty of Medicine and Dentistry, University of Valencia-INCLIVA, CIBERER, E-46010 Valencia, Spain; federico.v.pallardo@uv.es; 3Vall d’Hebron Institut de Recerca, E-08035 Barcelona, Catalunya, Spain; lyakhovich@gmail.com; 4Institute of Molecular Biology and Biophysics of the “Federal Research Center of Fundamental and Translational Medicine”, Novosibirsk 630117, Russia; 5Department of Life and Environmental Sciences, Polytechnical University of Marche, I-60100 Ancona, Italy; l.tiano@staff.univpm.it; 6Internal Medicine Unit, San Francesco d’Assisi Hospital, I-84020 Oliveto Citra (SA), Italy; m.fittipaldi@aslsalerno.it

**Keywords:** aging-related disorders, oxidative stress, mitochondria, mitochondrial dysfunction, mitochondrial nutrients, optic neuropathies, microbiome

## Abstract

A number of aging-related disorders (ARD) have been related to oxidative stress (OS) and mitochondrial dysfunction (MDF) in a well-established body of literature. Most studies focused on cardiovascular disorders (CVD), type 2 diabetes (T2D), and neurodegenerative disorders. Counteracting OS and MDF has been envisaged to improve the clinical management of ARD, and major roles have been assigned to three mitochondrial cofactors, also termed mitochondrial nutrients (MNs), i.e., α-lipoic acid (ALA), Coenzyme Q10 (CoQ10), and carnitine (CARN). These cofactors exert essential–and distinct—roles in mitochondrial machineries, along with strong antioxidant properties. Clinical trials have mostly relied on the use of only one MN to ARD-affected patients as, e.g., in the case of CoQ10 in CVD, or of ALA in T2D, possibly with the addition of other antioxidants. Only a few clinical and pre-clinical studies reported on the administration of two MNs, with beneficial outcomes, while no available studies reported on the combined administration of three MNs. Based on the literature also from pre-clinical studies, the present review is to recommend the design of clinical trials based on combinations of the three MNs.

## 1. Introduction

The multiple phases of mitochondrial function and the combined relationships of mitochondrial machineries with OS have been established since early studies in the mid-20th century and the ensuing studies are still a major subject of up-to-date reports [1,2,3]. Thus, mitochondria are commonly recognized as the main cell’s powerplant and the first victims of their energy-producing and prooxidant functions. These organelles also display strong antioxidant properties, and this is the case for ALA, involved in the citric acid cycle, for CoQ10, involved in the electron transport chain (ETC), and for CARN, the main carrier of acyl groups across the mitochondrial membrane. This background knowledge has led to the awareness of the key roles of mitochondrial cofactors in aging and in ARD. Further studies were focused on an extensive number of dysmetabolic and inborn disorders [4,5,6,7,8,9,10,11,12], also including the group of mitochondrial diseases dating back to the initial definition of “mitochondrial medicine” by Luft (1994) [6]. In turn, this awareness has generated a body of experimental and clinical studies targeted to the potential use of these mitochondrial cofactors, termed as “mitochondrial nutrients” (MNs) [13,14,15,16,17]. These mitoprotective strategies in counteracting ARD progression have been applied in a thriving body of in vitro and animal investigations, as well as in recent and current clinical trials, which are highlighted in the present review. The use of individual MNs in most of the published reports is critically discussed, in view of the as yet scanty reports using combinations of two MNs in clinical studies [18,19] or in experimental ARD models [20,21,22], in view of our suggested use of a MN “triad” (ALA + CoQ10 + CARN) in prospect study design.

## 2. Biochemical Principles and Prospects for the Use of Mitochondrial Nutrients in ARD

ALA is an organosulfur compound derived from octanoic acid, whose main chemical feature is the presence of two sulfur atoms bound by a disulfide bond. ALA is an essential cofactor in many enzyme complexes. It is especially important in the Krebs cycle for energy production by mitochondria and for pyruvate dehydrogenase activity [23].

ALA is metabolized in different ways when administered to mammals as a dietary supplement in various combinations; the safety of ALA has been demonstrated in numerous clinical studies [24,25,26]. A recent review by Dos Santos et al. [27] on the role of ALA in Alzheimer’s disease (AD) assessed the pharmacokinetic profile, bioavailability, therapeutic efficacy, safety, and effects of combined use with centrally acting drugs that demonstrate the mitochondrial mechanisms of ALA involved in AD protection.

Coenzyme Q identifies a family of lipophilic cofactors that are ubiquitous in many organisms [28]. The most abundant form in humans is CoQ10, which has a side chain of ten isoprenoid units. As the other MNs considered, it is an endogenous molecule, but it is also taken with food, and as such it is considered a dietary supplement. The main chemical characteristic of CoQ10 is its capability to act as an electron carrier. It plays a central role in the mitochondrial ETC, which is responsible for the transport of electrons from the respiratory complexes I (NADH dehydrogenase) and II (succinate dehydrogenase), to complex III (coenzyme Q: cytochrome c–oxidoreductase). In its reduced state, ubiquinol, it is also a free radical scavenger, so it can be considered as a physiological antioxidant. Ubiquinol protects biological membranes from oxidation, inhibits lipid peroxidation, and participates in α-tocopherol recycling.

CoQ10 levels were found decreased in patients affected by cerebellar multiple system atrophy (MSA) [29,30]. The authors suggested that CoQ10 deficiency may contribute to the pathogenesis of MSA, and that increasing CoQ10 levels by supplementation or upregulation of its biosynthesis may represent a novel treatment strategy for MSA patients. Analogous conclusions were provided by Shimizu et al. [31], and by Hargreaves and Mantle [32] that focused on the potential role of CoQ10 supplementation in the treatment of tissue fibrosis and cardiovascular risk, by improving cardiovascular function and reducing the risk of cardiovascular-associated mortality.

Besides the content, the oxidative status of CoQ10 is also noteworthy. CoQ10 can be present in oxidized (ubiquinone) or reduced (ubiquinol) forms. As mentioned, It should be noted that ubiquinol is the active antioxidant form of CoQ10 [33,34]; therefore, unlike ubiquinone, which requires activation of reductase activity to be chemically activated as an antioxidant, it provides direct efficacy, improved bioavailability, and, therefore, greater potential for treatment.

Moreover, a CoQ10 derivative, ubiquinone and ubiquinonyl-decyl-triphenylphosphonium (MitoQ) [35], exhibits the property of directly penetrating across the mitochondrial membrane. MitoQ has a recognized use as a CoQ10-based dietary supplement, as evaluated in experimental and clinical studies [36].

Carnitine (CARN, β-hydroxy-γ-N-trimethylaminobutyric acid, 3-hydroxy-4-N,N,N-trimethylaminobutyrate) is a quaternary ammonium compound associated with the transport of fatty acids in the mitochondrial membrane through CARN palmitoyltransferase I and CARN palmitoyltransferase II [37]. This process is of paramount importance for the delivery of free fatty acids released from different tissues, especially adipose tissue, to the mitochondria. These fatty acids will be used for ATP production after β oxidation. The CARN-mediated mitochondrial entry process is a rate-limiting factor for fatty acid oxidation and hence the production of cellular ATP.

## 3. Basic Research Studies

A body of literature has focused on the effects of MNs in aged animals, and in vivo as well as in vitro ARD models. As shown in Table 1, ALA has been tested in aged or ARD rat and canine models [21,22,38,39] such as AD and diabetic neuropathy, with beneficial effects on a number of endpoints. With one exception to these effects, ALA—unlike CoQ10—failed to improve nerve conductivity [21]. Three studies tested ALA and CoQ10 alone [21,22] or in combination [20], and two studies tested ALA with multiple antioxidant supplements [40,41]. Among these studies, Sadeghiyan Galeshkalami et al. [20] reported that the combination of ALA + CoQ10 successfully ameliorated the conditions of diabetic rats by preventing apoptosis and neuronal degeneration, regulating caspase 3 expression and inducing ATP production.

Several in vitro studies have reported the effects of ALA in rat cell cultures [41,42] and CoQ10 in human cells [43,44], consistently showing decreased ROS production and inflammation, as well as an increase in thiol production and an improvement in ETC. Polyphenols can exert antioxidant or prooxidant effects depending on the oxygen and glucose levels used in the cell culture. In addition, high concentrations of glucose in the cultured cell medium sometimes result in a glycolitic switch (Warburg effect).

Altogether, it can be assumed that most of the literature on animal and in vitro models can predict analogous beneficial effects of ALA and CoQ10—or a combination of both [19,20]—in people with ARD.

## 4. Mitochondrial Nutrient-Based Clinical Trials in Aging-Related Disorders

An extensive body of literature has been devoted to clinical trials testing MNs in ARD patients, focused on CVD, T2D, and other ARD, as shown in Table 2.

Among trials conducted in patients with CVD, CoQ10 was tested alone or with antioxidant supplements [45,46,47,48,49] and beneficial effects included decreased fibrosis, morbidity, and mortality, as well as decreased levels of pro-inflammatory cytokines, with improved left ventricular function and quality of life. A clinical trial conducted in CVD patients by McMackin et al. [18], testing a combination of ALA and acetyl-CARN, reported a significant improvement of blood pressure and vascular tone. Two other clinical trials [50,51] were conducted on acyl-CARN in CVD patients, which also reported positive effects on CVD endpoints, as well as OS parameters such as decreased concentrations of malondialdehyde and 4-hydroxynonenal. Altogether, one may recognize that CVD patients may benefit from adjuvant treatments with CoQ10, ALA, and acyl-CARN [18,45,46,47,48,49,50,51].

An extensive body of research evidence from the 1960s to the 1990s established a protective role for ALA in relation to diabetic complications and progression [24,52,53,54,61,62,63,64], up to recognition by the German Drug Agency in 1994 [65] as a prescription agent in treating diabetic patients.

Several clinical trials were carried out in patients affected by other ARD. Shinto et al. [55] tested the effects of ALA + ω-3 fatty acids in patients with AD, and reported an improvement in neurological conditions. Galasko et al. [66] found improvement in cerebrospinal fluid biomarkers in AD patients receiving ALA, CoQ10, and antioxidant vitamins. Improvement in inflammatory markers was found by Sanoobar et al. [67] in patients with multiple sclerosis following treatment with CoQ10. A report by Pistone et al. [68], testing CARN in elderly subjects with rapid muscle fatigue, found favorable effects on fatigue and serum lipids, as well as a decrease in total fat mass and an increase in total muscle mass. Administration of acetyl-CARN or propionyl-CARN reduced mental fatigue [69], while CoQ10 deficiency was found in patients with chronic fatigue syndrome (CFS) [70]. An early study reported on 30 patients with CFS to compare CARN and amantadine, showing a statistically significant clinical improvement in 12 out of the 18 CSF parameters after eight weeks of treatment [57]. A randomized, double-blind placebo-controlled trial on 73 CFS patients showed that administration of CoQ10 (200 mg/day) plus NADH (20 mg/day) conferred benefits on fatigue and biochemical parameters [58]. The on-going trial by Stough et al. [59] is investigating the effects of CoQ10 administration on cognitive and cardiovascular function, and on OS in elderly subjects.

Apart from the successful clinical trials of MNs in patients with ARD, other studies have focused on disorders of a different nature, yet variously related to mitochondrial anomalies. This was the case for the studies of MNs in Down syndrome [71,72], an example of an accelerated aging disease, fibromyalgia [60], and mitochondrial disorders [73,74], which reported suggestive evidence for MN-induced beneficial effects. Other clinical trials failed to report clear positive effects of MN administration in patients with some ARD, such as rheumatoid arthritis [75] or Parkinson’ disease [76], or of acetyl-CARN in T2D patients [77].

Another example of age-related changes that implies medical relevance in dermatology is given by skin aging and measures to reduce the progression of wrinkles. In this area, a pre-existing—and current—literature has focused on the role of mitochondria in creating a prooxidant state since the early studies [78,79]. Hence, a line of studies has pursued MN-containing preparations [80,81], and a body of literature has reported the favorable effects of topical MN administration using ALA [82,83] or CoQ10 [84,85] or CARN [86], as evaluated in clinical trials. In several studies, MN administration was accompanied by other ingredients.

## 5. The Case of Progerias

Progerias, also termed progeroid syndromes (PGS), are a set of genetic disorders with various gene defects, yet exhibiting phenotypes characterized by symptoms of early aging, along with the involvement of a prooxidant state and MDF. The most investigated PGS are Hutchinson–Gilford progeria syndrome (HGPS), Cockayne syndrome (CS), Werner syndrome (WS), and Down syndrome (DS), altogether resulting in a number of signs of early aging [87,88,89].

Rivera-Torres et al. [90] studied fibroblasts from healthy subjects and patients with HGPS, as well as a mouse HGPS model, and found a marked downregulation of mitochondrial oxidative phosphorylation proteins, accompanied by MDF in HGPS cells. MDF has also been found in fibroblasts of adult progeroid mice expressing progerin or prelamin A. Tissue analysis of these mouse models has shown that the damaging effect of these proteins on mitochondrial function is time- and dose-dependent.

WS is recognized as a model of human aging. WS-affected patients display late onset (adolescence) with some of the characteristics of normal aging such as cataracts, hair graying, skin aging, and cancer propension [91]. In addition, WS individuals have a high incidence of inflammatory diseases such as atherosclerosis and T2D. The in vivo phenotype of WS is associated with significantly increased prooxidant endpoints in circulating leukocytes compared to other genetic diseases. These parameters included: (a) leukocyte and urinary 8-hydroxy-2′-deoxyguanosine; (b) blood glutathione; (c) plasma glyoxal and methylglyoxal; and (d) some plasma antioxidants (uric acid and ascorbic acid) [92]. The WRN mouse model was investigated by Lebel et al. [93], who found ascorbate-induced improvement in metabolic abnormalities in WRN mutant mice, such as reduced OS in liver and heart tissues and reversal of hypertriglyceridemia, hyperglycemia, and insulin resistance.

CS is an autosomal recessive segmental PGS with a complex series of pathological features. There are five genes responsible for this syndrome: CSA, CSB, XPB, XPD, and XPG, in which various mutations have been found [94]. The phenotypic effects of these mutations include a defective DNA oxidative damage repair system.

The consequences of OS and MDF in DS have been investigated from early studies and to the present. Ultrastructural abnormalities in cultured cerebellar neurons from trisomy 16 (Ts16) mice, a model of DS, have been reported by Bersu et al. [95]. The abnormalities included, among others, irregular-shaped mitochondria and decreased microtubule levels, which were not observed in any of the control neurons. Druzhyna et al. [96] reported defective repair of mitochondrial DNA (mtDNA) oxidative damage in fibroblasts from DS patients, while Schuchmann and Heinemann [97] found mitochondria-associated anomalies in neurons of Ts16 mice. A selective decrease in respiration was detected with the Complex I substrates, malate and glutamate, but not with the succinate of the Complex II substrate, in isolated cortex mitochondria of the Ts16 mice [98]. Conti et al. [99] investigated the gene expression profile of chromosome 21 (Hsa21) using oligonucleotide microarrays in the hearts of human fetuses with and without Hsa21 trisomy, and found that Hsa21 gene expression in trisomic samples was increased by 1.5 times. Evaluation of functional classes and analysis of the enrichment of a set of 473 genes revealed a downregulation of genes encoding mitochondrial enzymes and an upregulation of genes encoding extracellular matrix proteins. The authors concluded that the dose-dependent upregulation of Hsa21 genes causes dysregulation of genes responsible for mitochondrial function and organization of the extracellular matrix in the fetal heart of trisomic subjects [99]. Overall, the body of evidence in PGS and in DS points to a central role of OS and MDF in these disorders, prompting the design of mitoprotective clinical strategies to mitigate the adverse phenotypic effects of PGS [100].

## 6. The Case of Optic Neuropathies

Although MNs have been studied for their ability to decelerate the aging of various organisms, the effectiveness in preventing ARD must be weighed against the optimal dose and number of drugs. Therefore, the search for a suitable disease model that allows obtaining an unbiased readout and monitoring clinical outcomes can greatly strengthen the applicability of MNs. Since aging likely contributes to the vulnerability of the optic nerve over time, progressive optic neuropathy in the form of glaucoma may become such a disease. Indeed, treatment of glaucoma with MNs may provide some clear data for monitoring clinical improvements. The rationale and efficacy of CoQ10 in protecting from retinal diseases and neuroretinal degeneration were reviewed by Zhang et al. [101,102]. CoQ10 can ameliorate OS-mediated alterations in mitochondria of the optic nerve head astrocytes [103]. Intraocular administration of CoQ10 prevents retinal damage in rats [104,105], inhibits glutamate excitotoxicity and OS-mediated changes in mitochondria of a mouse model of glaucoma [106], and demonstrates mitochondrial-mediated neuroprotection in a rodent model of ocular hypertension [107]. In a study of 43 patients with open-angle glaucoma associated with CoQ10 and vitamin E supplementation, Parisi et al. [56] found improvement in retinal function followed by an increase in visual cortical responses. The first randomized, double-blind controlled-fashion study in 612 individuals with glaucoma is currently underway to investigate the neuroprotective effects of CoQ10 and Vit E [108]. For another MN, CARN, Calandrella et al. [109] reported its neuroprotective, antiapoptotic, and antioxidant effects in a glaucoma rat model. In a randomized double-blind pilot study of 40 patients with ocular discomfort syndrome secondary to glaucoma, treatment for 15 days with eye drop solutions containing eledoisin and CARN showed a 50% reduction in symptom severity [110]. Finally, ALA treatment limits glaucoma-related retinal ganglion cell death and dysfunction [111] and exerts a neuroprotective effect against OS in retinal neurons in vitro and in vivo [112]. In addition, topical administration of ALA to rabbit eyes prevents and/or reduces fibrosis by inhibiting the pathways of inflammation, revascularization, and accumulation of the extracellular matrix [113]. However, it should be taken into account that the etiology of glaucoma is extensive, and the use of MNs can be recommended for those glaucoma-related processes where mitochondria are involved in the physiopathological process.

## 7. MNs in the Content of Microbiome and Aging

Given the recent advances in metabolomics, the gut microbiome is currently viewed as an endocrine organ that produces a wide range of metabolites in response to foods. In this regard, all mitochondrial supplements and the gut microbiome constitute the food metabolome, obtained as a result of the digestion and biotransformation of food [114,115]. Changes in the gut microbiome are associated with a variety of diseases, including neurological disorders, CVD [116], and metabolic diseases such as obesity and T2D [117]. The aging process also affects the food metabolome and its interaction with the immune system [118]. In this regard, the MNs produced by the microbiome can be exchanged between kingdoms with mammalian cells, thus altering or enhancing the need for energy production in some mitochondrial disorders. For instance, in a murine cell culture model of the central nervous system, the structural analogs of CARN produced by anaerobic commensal bacteria in the intestine can colocalize with CARN in the brain and interfere with CARN-mediated fatty acid oxidation [119].

CARN transports long-chain fatty acids across the mitochondrial membrane for energy production and promotes the excretion of toxic organic acids in urine [120]. Some age-related metabolites (creatinine, creatine, glycine, betaine/trimethylamine N-oxide, citrate, succinate, and acetone) can be cleared from the body over a long period of time, which suggests their use in patients with mitochondrial disorders [121,122]. The role of CoQ10 in the metabolome is still unclear, since it is difficult to separate ubiquinone synthesized in eukaryotic cells from intestinal menaquinone in bacterial cells [123]. However, recent reports provided evidence that CoQ10 supplementation affects the metabolomic profile in the urine of elderly people and may be beneficial for healthy aging [124].

It is noteworthy that four alternative forms of quinone biosynthetic enzymes have been identified in the human intestinal microbiome [123]. Disruption of quinone biosynthesis can lead to metabolic disorders, since all these quinones are growth factors for the human intestinal microbiota [124]. Hence, substitution with an MN mix can compensate for the altered microbiome, leading to clinical improvement of the whole organism. In turn, ALA has a direct effect on lipid metabolism and the liver, modulating mitochondrial function [125], and can reinforce the redox state in mice, which leads to modulation of intestinal microbiota and improved metabolic health [126].

Although data on the metabolism and pharmacokinetics of MNs in humans are limited, several studies have shown that withdrawal of MN (CoQ10) supplementation returns levels to normal within a few days [25]. Therefore, novel MN formulations should be developed to increase or prolong the bioavailability of MNs. Given the importance of the MN in maintaining the intermetabolic pathways of the microbiome, serious consideration should be given to the maladjustment of the intestinal and host microbiota to correct age-related dysbiosis.

## 8. Aging, Antioxidants, and Possible Advantages of the MN “Triad”

In view of the concept of ROS-mediated ARD, a number of antioxidants have been proposed as a front line treatment for aging. Without diminishing the benefits of the role of antioxidants as anti-aging compounds, it should be emphasized that there is still no evidence base for the use of such compounds in medical practice. Furthermore, no evidence for the accumulation of commonly used gerontoprotectors (e.g., *tert*-butylhydroxylamin) or vitamin E in mitochondria has been reported so far. In this regard, the concept of preventing an age-related increase in the level of mitochondrial ROS should prompt ad hoc investigations [127]. These may lead to search for and to use substances aimed not at the general elimination of any ROS in cells and tissues, but at maintaining a low ROS content in the intramitochondrial space. Although an excess of mitochondrial ROS is detrimental for any organism, ROS are required both for the immune system against pathogens and for redox signaling [128]. Recently, mild depolarization of the inner mitochondrial membrane was shown as a major component of an anti-aging program [129]. ROS perform a physiological function as messenger molecules, but they also constitute reactive molecules potentially harmful for the cellular system. In particular, mitochondria are an important source and target of ROS and are critical modulators of the cellular metabolic process and programmed cell death.

In this context, the proposed MNs may be the safest triad for mild mitochondrial effects. Lipoic acid can directly scavenge superoxide and improve mitochondrial function [130], CoQ10 is an important endogenous lipid-soluble antioxidant [131], and CARN reduces OS in many biological systems [132,133]. Despite the fact that the components of the MN triad are safe for use in clinical trials and have been studied for their ability to decelerate the aging of different organisms, their effectiveness will most likely depend on the selection of the optimal doses and on the time of their administration.

## 9. Concluding Remarks

Beyond the above-cited experimental and clinical reports, the extensive literature discusses the potential benefits of treating ARD and mitochondria-defective diseases using MNs [14,25,134,135,136,137,138,139,140,141,142,143,144]. Consistent with the available database from pre-clinical and clinical investigations, the available MDF-related reviews mainly focus on the potential advantages of using individual MNs in the treatment of mitochondria-defective disorders, with the exception of the “mitochondrial cocktail” perspective proposed by Tarnopolsky [136] and as discussed in our previous review [137].

One might argue that an approach based on individual MNs may contradict basic, well-established knowledge about mitochondrial function based on the combined synergism of different metabolic systems, as was built by Palade in 1964 [2] and in an extensive body of basic research over sixty years. It is postulated that the regulation of mitochondrial homeostasis by any of the MN components during aging affects age-related declines. Thus, the choice of treatment for MDF disorders such as ARD, CVD, and T2D or neurodegenerative diseases with only one MN may be justified in the case of precise information about a deficiency in one specific MN. An alternative approach may consist of pharmaco-epidemiological studies aimed at addressing the potential impact of MNs on the course of ARD. This research might both provide an important contribution to the evidence-based need for rational decision making in medicine and reduce the financial burden of full-scale clinical trials. The knowledge gained from such research will significantly expand the therapeutic strategies available for treating ARD and help reduce the enormous public health burden of these diseases world-wide. It can be assumed that a line of investigations may be warranted in an attempt to recognize a specific deficiency in a given MN that would justify treatment with this defective MN. In the absence of this individual MN deficit, a temporary solution in study design could be based on the following steps:(1)Preclinical studies (lifelong treatments and/or experimental diabetes) comparing the protective effects of CoQ10 vs. ALA vs. (acyl-)CARN and combinations thereof;(2)Clinical trials (morbidity, mortality, and cognitive decline; T2D compensation), which may be conducted following the results of pre-clinical investigations.

The physiological roles of the three MNs, along with the outcomes of several reports, suggest the expected safety of treatments with MN combinations, thereby safeguarding ethical issues in research proposals.

Preclinical studies are recognized to feature year-long durations, similar to investigations to test chronic toxicity or carcinogenesis in rodents, with a life expectancy of 2 years. Potential results from animal investigations could lay the groundwork for the design of clinical trials that can be multi-year- or 10-year-long in accordance with, for example, the guidelines of the National Social Life, Health and Aging Project [145]

Expecting broader and more targeted studies, we can at least recommend the long-term management of elderly patients and patients with ARD through chronic treatment with appropriate dosages and combinations of MNs and physical exercise, not excluding specific treatment methods aimed at the disease.

## Figures and Tables

**Table 1 ijms-21-07060-t001:** Main effects of mitochondrial cofactors (ALA, CoQ10, and CARN) in aging-related animal and in vitro models.

Models	Cofactor	Reported Effects	References
Aged and young rats	ALA	regulation of hepatic genes associated with lipid and energy metabolism and circadian rhythm	[38]
Aged and young rats	ALA + other supplements	decreased IL-1β, IL-6, and TNF-α levels	[39]
Canine model of human	ALA + other	improvement in spatial attention	[40]
Rat diabetic neuropathy	ALA + CoQ10	prevented apoptosis and neuron degeneration regulated expression of caspase 3 and induced ATP	[20]
Rat diabetic neuropathy	ALA or CoQ10	CoQ10, not ALA, improved nerve conductivity	[21]
Aged dogs	ALA + CARN	improved social, cognitive, and physical activity	[22]
Rat pancreatic cells	ALA	suppressed p38 and p53 genes, reducing ROS formation and enhancing thiol levels	[41]
Rat embryonic fibroblasts	ALA	decreased ROS formation and inflammation, and affected cell division	[42]
Human cells from skin biopsies	CoQ10	improved electron transport chain and its antioxidant potential	[43]
Human dermal fibroblasts	CoQ10	CoQ10 decrease lowers mitochondrial permeability and results in bioenergetic dysfunction	[44]

**Table 2 ijms-21-07060-t002:** Selected clinical studies testing the effects of ALA, CoQ10, or (acyl-) CARN in aging-related disorders.

Diseases	Cofactor	Reported Effects	References
**1.Cardiovascular Mortality**			
Cardiovascular mortality	CoQ10 + Se	Decreased fibrosis	[45]
Chronic heart failure	CoQ10	Decreased morbidity and mortality	[46]
Chronic heart failure	CoQ10 + creatine	Increased peak oxygen consumption	[47]
	CoQ10 + supplements	Improved left ventricular function, levels of pro-inflammatory cytokines, and quality-of-life	[48]
Coronary artery disease	CoQ10	Increases antioxidant enzyme activity	[49]
	ALA + acetyl-CARN	Improved regulation of blood pressure and vascular tone	[18]
Increased cardiovascular risk	acetyl-CARN	Ameliorating hypertension and insulin resistance	[50]
Peripheral artery disease	propionyl-CARN	Decreased plasma malondialdehyde and 4-hydroxynonenal concentrations, and the plasma nitrite/nitrate ratio	[51]
**2. Type 2 Diabetes Mellitus**			
	polarized light +	Decreased lactate dehydrogenase activity	[19]
	ALA + CoQ10 +	Decreased plasma lipid peroxides	
	α-tocopherol		
	ALA	Improved glycemic status and lipid peroxidation	[52]
	ALA	Decreased oxidative stress, blood glucose, and lipid levels	[53]
	ALA	Improved diabetic polyneuropathy	[54]
**3. Miscellaneous Disorders**			
Alzheimer disease	ALA + ω-3 fatty acids	Improved mental and cognitive state	[55]
Glaucoma	ALA	Decreased retinal ganglion cell death and dysfunction	[56]
Multiple sclerosis	CoQ10	Improved inflammatory markers	[57]
Elderly rapid muscle fatigue	CARN	Reduction in total fat mass, increase in total muscle mass, and favorable effects on fatigue and serum lipids	[58]
Age-related cognitive decline	CoQ10	On-going study of cardiovascular function, oxidative stress, liver function, and mood	[59]
Fibromyalgia	CoQ10	Significant improvement in clinical and headache symptoms	[60]

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
