# Peer review of "Aging-Related Disorders and Mitochondrial Dysfunction: A Critical Review for Prospect Mitoprotective Strategies Based on Mitochondrial Nutrient Mixtures"

_ijms, 2020, doi:10.3390/ijms21197060_

Round 1

Reviewer 1 Report

This is a very well written review of the MN and diseases that could be affected or improved with supplementation of specific Mn in ARD. This information is important and clinically relevant. Im glad to see this review and believe it will help to move the field forward in regard to mt function in pathophysiological conditions.

This sentence is confusing and I don’t understand what the authors are trying to say

“It should be noted that ubiquinol is recognized as a form of direct action, as it was established from the first studies [33,34], in contrast to ubiquinone, which requires reductase activity.”

I think it would flow better if “A CoQ10 derivative, ubiquinone and ubiquinonyl-decyl-triphenylphosphonium (MitoQ) [35] exhibits the property of direct penetrating across mitochondrial membranes and has received recognized use as CoQ10-based dietary supplement as evaluated in experimental and clinical studies [36].” Came before it.

Ofadult needs to be corrected on page 5

Andorganization on page 6

model,of

may be eliminated from the body after CARN has been taken for an extended period of time, suggesting its use in patients with mitochondrial disorders, what do the authors mean, suggesting benefits or what for mn disorders?

“Although the data on the metabolism and pharmacokinetics of MNs in humans are limited, several studies have shown that withdrawal of MN (CoQ10) supplementation returns levels to normal within a few days [25]. Therefore, novel MN formulations should be developed to increase or prolong bioavailability of MNs. Considering the importance of the MN diet in the maintenance of intermetabolic pathways of the microbiome, one should seriously consider the issue of maladjustment of the intestinal microbiota and the host for the correction of age-related dysbiosis.” This section doesn’t have the same font size or spacing as the rest of the review.

deserves all attention, I think all can be removed

search for and use substan… there is a word missing perhaps of

Too high ROS should be reworded, not very scientific

and further basic research during sixty years this phrase is really unnecessary and there are no citations following the mention of basic science so it probably could be removed all together.

basedon the;  page 8

This is a very well written review of the MN and diseases that could be affected or improved with supplementation of specific Mn in ARD. This information is important and clinically relevant. Im glad to see this review and believe it will help to move the field forward in regard to mt function in pathophysiological conditions.

This sentence is confusing and I don’t understand what the authors are trying to say

“It should be noted that ubiquinol is recognized as a form of direct action, as it was established from the first studies [33,34], in contrast to ubiquinone, which requires reductase activity.”

I think it would flow better if “A CoQ10 derivative, ubiquinone and ubiquinonyl-decyl-triphenylphosphonium (MitoQ) [35] exhibits the property of direct penetrating across mitochondrial membranes and has received recognized use as CoQ10-based dietary supplement as evaluated in experimental and clinical studies [36].” Came before it.

Ofadult needs to be corrected on page 5

Andorganization on page 6

model,of

may be eliminated from the body after CARN has been taken for an extended period of time, suggesting its use in patients with mitochondrial disorders, what do the authors mean, suggesting benefits or what for mn disorders?

“Although the data on the metabolism and pharmacokinetics of MNs in humans are limited, several studies have shown that withdrawal of MN (CoQ10) supplementation returns levels to normal within a few days [25]. Therefore, novel MN formulations should be developed to increase or prolong bioavailability of MNs. Considering the importance of the MN diet in the maintenance of intermetabolic pathways of the microbiome, one should seriously consider the issue of maladjustment of the intestinal microbiota and the host for the correction of age-related dysbiosis.” This section doesn’t have the same font size or spacing as the rest of the review.

deserves all attention, I think all can be removed

search for and use substan… there is a word missing perhaps of

Too high ROS should be reworded, not very scientific

and further basic research during sixty years this phrase is really unnecessary and there are no citations following the mention of basic science so it probably could be removed all together.

basedon the;  page 8

Author Response

Also on behalf of my co-authors, I express my sincerest thanks for your pertinent corrections, which were determinant for improving details and overall quality of our manuscript.

Rev 1

1) This sentence is confusing and I don’t understand what the authors are trying to say

“It should be noted that ubiquinol is recognized as a form of direct action, as it was established from the first studies [33,34], in contrast to ubiquinone, which requires reductase activity.”

I think it would flow better if “A CoQ10 derivative, ubiquinone and ubiquinonyl-decyl-triphenylphosphonium (MitoQ) [35] exhibits the property of direct penetrating across mitochondrial membranes and has received recognized use as CoQ10-based dietary supplement as evaluated in experimental and clinical studies [36].” Came before it.

The sentence was made clearer, with the following revision:

CoQ10 levels were found decreased in patients affected by cerebellar multiple system atrophy (MSA) [29,30]. The authors suggested that CoQ10 deficiency may contribute to the pathogenesis of MSA, and that increasing CoQ10 levels by supplementation or upregulation of its biosynthesis may represent a novel treatment strategy for MSA patients. Analogous conclusions were provided by Shimizu et al. [31], and by Hargreaves and Mantle [32] that focused on the potential role of CoQ10 supplementation in the treatment of tissue fibrosis and cardiovascular risk, by improving cardiovascular function and reducing the risk of cardiovascular-associated mortality.

Besides the content, the oxidative status of CoQ10 is also noteworthy. CoQ10 can be present in the oxidized (ubiquinone) or reduced (ubiquinol) forms. As mentioned, it should be noted that ubiquinol is the active antioxidant form of CoQ10 [33,34]; therefore, unlike ubiquinone, which requires activation of reductase activity to be chemically activated as an antioxidant, it provides direct efficacy, improved bioavailability and therefore, greater potential for treatment.

Moreover, a CoQ10 derivative, ubiquinone and ubiquinonyl-decyl-triphenylphosphonium (MitoQ) [35] exhibits the property of directly penetrating across the mitochondrial membrane. MitoQ has a recognized use as CoQ10-based dietary supplement, as evaluated in experimental and clinical studies [reviewed in 36].

2) "Ofadult" needs to be corrected on page 5

CORRECTED

3) Andorganization on page 6

CORRECTED

4) model,of

CORRECTED

5) may be eliminated from the body after CARN has been taken for an extended period of time, suggesting its use in patients with mitochondrial disorders, what do the authors mean, suggesting benefits or what for mn disorders?

In order to better substantiate this statement, ref. #121 is now changed to the more appropriate paper by Vallance et al. (2018), linking the use of CARN in mitochondrial disorders.

6) “Although the data on the metabolism and pharmacokinetics of MNs in humans are limited, several studies have shown that withdrawal of MN (CoQ10) supplementation returns levels to normal within a few days [25]. Therefore, novel MN formulations should be developed to increase or prolong bioavailability of MNs. Considering the importance of the MN diet in the maintenance of intermetabolic pathways of the microbiome, one should seriously consider the issue of maladjustment of the intestinal microbiota and the host for the correction of age-related dysbiosis.” This section doesn’t have the same font size or spacing as the rest of the review.

CORRECTED

7) deserves all attention, I think all can be removed

CORRECTED

8) search for and use substan… there is a word missing perhaps of

Revised as:

In this regard, the concept of preventing an age-related increase in the level of mitochondrial ROS should prompt ad hoc investigations [127]. These may lead to search for and to use of substances aimed not at the general elimination of any ROS in cells and tissues, but at maintaining a low ROS content in the intramitochondrial space.

9) Too high ROS should be reworded, not very scientific

CORRECTED

10) and further basic research during sixty years this phrase is really unnecessary and there are no citations following the mention of basic science so it probably could be removed all together.

CORRECTED as:

built by Palade in 1964 [2] and in an extensive body of basic research during sixty years.

11) basedon the;  page 8

CORRECTED

Reviewer 2 Report

The review “Aging-related Disorders and Mitochondrial Dysfunction: A Critical Review for Prospect Mitoprotective Strategies Based on Mitochondrial Nutrient Mixtures” by Giovanni Pagano et al. focuses on promising mitochondrial strategies based on mixtures of mitochondrial nutrients in aging-related disorders. As mitochondrial nutrients, the authors consider a mixture of substances such as α-lipoic acid, Coenzyme Q10, and carnitine. The authors cite both fundamental and preclinical investigates and a recommended clinical trial design based on combinations of the three mitochondrial nutrients. This review is well described and accessible to any reader. The problem raised in the review is important and therefore the described material should be published in the journal International Journal of Molecular Sciences.

Author Response

Dear Madam/Sir,

thank you very much for your appreciation of my colleagues' and my manuscript.

Warm regards,

Giovanni Pagano